# Community composition of microbial microcosms follows simple assembly rules at evolutionary timescales

Nittay Meroz [1✉], Nesli Tovi[1], Yael Sorokin[1] & Jonathan Friedman [1✉]

Managing and engineering microbial communities relies on the ability to predict their composition. While progress has been made on predicting compositions on short, ecological timescales, there is still little work aimed at predicting compositions on evolutionary timescales. Therefore, it is still unknown for how long communities typically remain stable after reaching ecological equilibrium, and how repeatable and predictable are changes when they occur. Here, we address this knowledge gap by tracking the composition of 87 two- and three-species bacterial communities, with 3–18 replicates each, for ~400 generations. We find that community composition typically changed during evolution, but that the composition of replicate communities remained similar. Furthermore, these changes were predictable in a bottom-up approach—changes in the composition of trios were consistent with those that occurred in pairs during coevolution. Our results demonstrate that simple assembly rules can hold even on evolutionary timescales, suggesting it may be possible to forecast the evolution of microbial communities.

[1] Department of Plant Pathology and Microbiology, The Hebrew University of Jerusalem, Rehovot, Israel. ✉email: nittay.meroz@mail.huji.ac.il; yonatan.friedman@mail.huji.ac.il

Microbes are engines of biogeochemical cycles[1], agents of health and illness[2], and key players in food and biotechnological industries[3]. They are seldom found in a biotic void but rather form complex and diverse communities of multiple interacting species. The ongoing realization of the significant role microbial communities play in various environments[4–6], along with the potential that lies in the ability to manipulate and design them, has motivated research aiming to disentangle the rules that govern microbial ecology[7].

The ability to rationally design or manipulate communities depends on our ability to predict their composition. Currently, predicting the structure and dynamics of microbial communities commonly relies on fitting specific ecological models, quantitative mechanistic models, or using genome-scale metabolic models[8–11]. Challenges in fitting such models[11] and concerns regarding their validity[12] have recently led us to develop an alternative non-parametric predictive framework, which was validated using laboratory experiments in microbial microcosms[13,14], and in the gut of *C. elegans*[15]. In this bottom-up framework, the composition of multispecies communities is inferred using the measured pairwise interactions between the members of the community. However, while this approach and other notable frameworks had made progress, none of these take into account changes in species abundances that could occur during evolution; thus, the timescales at which these frameworks remain predictive are still unclear.

Understanding the dynamics of community composition over evolutionary timescales is crucial for designing sustainable microbial communities; adaptations that occur while species coevolve as a community, could potentially result in loss of diversity or lead to loss of desired functions[16]. Experiments involving microbial communities have demonstrated that changes in interspecies interactions and metabolic activities are common when species evolve in a community, and are often different than those that occur when species evolve in isolation[17–24]. Yet we still lack a clear understanding of how prevalent such changes are, and how they affect community composition.

Evolutionary changes in community composition may not be repeatable, making them challenging to predict. At ecological timescales, community composition is typically highly deterministic—repeated experiments converge to similar compositions from the identical initial conditions[13]. Many features of single-species evolution are also often found to be strikingly repeatable[25–34], however, it is not clear whether similar repeatability occurs when species coevolve within communities. A recent experimental study found that the composition of 96-replicate communities composed of six species of soil bacteria was significantly altered during coevolution, and that replicates clustered into four distinct types that varied in species abundances[35]. In another study that measured the dynamics of a three-species multitrophic microbial community for three months, variation between replicates remained strikingly low for the duration of the experiment[36]. How coevolution affects the predictability of community-level features, including community composition, is still an under-explored question.

Here, we use experimental evolution of 87 two- and three-species bacterial communities, with 3-18 replicates each, to test the prevalence, repeatability, and predictability of changes in community composition over ~400 generations of coculturing. We find that over these timescales community composition typically changes significantly, and that variability between replicate communities typically increases. However, replicate communities remain similar enough to allow predictions to be made. Specifically, we show that a simple assembly rule based on the outcomes of pairwise coculturing, originally developed for ecological timescales[13], can be expanded for evolutionary

timescales. In contrast, we find that predictions based on species' growth parameters, which have reasonable accuracies at ecological timescales, have no predictive power at evolutionary timescales.

## Results

We conducted a high-throughput evolutionary experiment, propagating 87 two- and three-species communities for ~400 generations in well-mixed microcosms containing M9 minimal-media supplemented with three carbon sources—galacturonic acid, acetate, and serine (Fig. 1A–C, "Methods"). All communities consist of subsets of a library of 16 heterotrophic soil bacterial species (Supplementary Table 1). Multiple replicates (3–18, Supplementary Data 1) of each community, as well as of the 16 monocultures, were cultured in 200 µl minimal media and passaged through 38 cycles of 48 h of growth and 1500-fold dilution into fresh media (Fig. 1, Methods). The relative abundances of each species within communities were measured at 9 timepoints throughout the experiment by plating and counting colonies, which are morphologically distinct for each species (Fig. 2A, Supplementary Figs. 1, 2). To increase the chance that the communities included in the evolution experiment consist of species that coexist on evolutionary timescales, we only included pairs that were found to coexist for ~60 generations in a preliminary experiment, and trios composed of such pairs (Methods). In most communities all species survived throughout the evolution experiment (>74% of all communities; >83% of pairs and >65% of trios).

**Community composition typically changes over evolutionary timescales.** The first few dozen generations are characterized by significant shifts in community composition (Fig. 2A,B, Supplementary Fig. 4A). These rapid changes are consistent with previous studies[13,37], and likely reflect a transition from the arbitrary initial abundances that species were inoculated at to an ecological equilibrium. This is supported by the fact that over these timescales, pairs inoculated at varying initial fractions converge to similar compositions (Supplementary Fig. 3), and that changes decelerate considerably after ~50-~70 generations (Fig. 2B, Supplementary Fig. 4A), suggesting that these timescales were sufficient for most communities to reach an ecological equilibrium.

Compositions reached at ecological timescales typically change during coevolution. Although the rate of change in community composition drops, communities continue to change slowly (Fig. 2B, Supplementary Figs. 4A, 5). These alterations accumulate such that as time goes by most communities diverge significantly from their ecological states (Fig. 2C, Supplementary Fig. 4B). While most communities change significantly, some communities remain strikingly stable throughout the entire experiment (Fig. 2A,C, Supplementary Figs. 1, 2, 4B), but it is still not clear what features contribute to their stability. These results underscore our limited ability to extrapolate community composition to evolutionary timescales from ecological data, and stress the need to adjust prediction rules for these timescales.

Community composition changes over evolutionary timescales due to heritable alterations in species' phenotypes. In order to evaluate whether compositional changes that occur on longer timescales are caused by evolutionary changes or by long-term ecological dynamics, we re-isolated strains from 30 pairs that evolved for ~400 generations. Pairs of evolved strains, as well as ancestral strains, were then cocultured again for ~53 generations from the same initial conditions used to initiate the evolution experiment. We observed that while ancestral strains reproduced the compositions reached at ecological timescales during the evolutionary experiment, evolved strains reached significantly

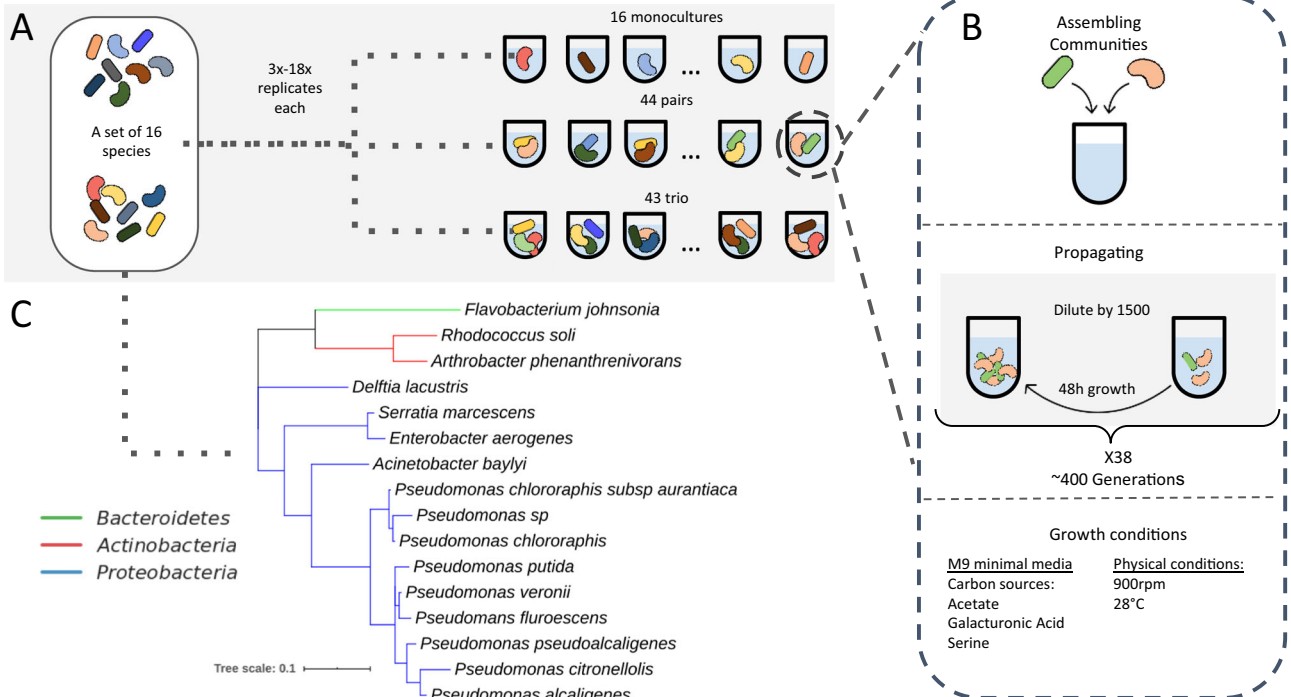

**Fig. 1 A high-throughput evolutionary experiment involving 87 communities composed of subsets of a 16 species set. A** Subsets of the 16 species were used to assemble 44 pairs and 43 trios. All 16 species were evolved in monocultures in parallel to the assembled communities. Each community was grown in 3–18 replicates, a full list of communities and the number of replicates is found in Supplementary Table 2. **B** Each community was grown in M9 minimal media supplemented with three carbon sources in cycles of 48 h growth-dilution by a factor of 1500. Overall, communities grew for 38 cycles, which correspond to ~400 generations. **C** Phylogenetic tree based on the full-length 16S sequence of the 16 species used for this study.

different compositions (Supplementary Figs. 6, 7, Methods), indicating that many compositional changes were evolutionary-driven. Next, we wanted to assess how repeatable are such evolutionary-driven compositional changes across replicates in all of our evolved communities.

**Replicate communities maintain similar compositions during coevolution.** While the variability in composition between replicate communities increases during coevolution, they remain more similar to each other than to random communities. As expected at ecological timescales, the rapid changes in community composition that occurred during the first ~70 generations were typically highly repeatable, with low variability between replicates. Subsequent changes were less repeatable and often resulted in increased variability between replicate communities (Fig. 3A, Supplementary Fig. 8A). Most communities (74%), had more replicate-to-replicate variability at generation ~400 than at generation ~70 (Fig. 3B, Supplementary Fig. 8B). Nonetheless, in most communities replicates remained more similar to each other than expected by chance throughout the experiment (Mann–Whitney $p$ value $\sim 8 * 10^{-19}$ at generation ~400, 87 communities, Supplementary Fig. 8A, "Methods"). Moreover, the species whose abundance increased by the largest factor during coevolution tended to be conserved across replicate communities (permutation test $p$ value $< 5 * 10^{-4}$, 87 communities; Fig. 3C, Supplementary Fig. 8C, "Methods"). These results reveal that although coevolution has a diverging effect on community composition, post-ecological alterations display some level of determinism and are thus potentially predictable.

**Pair compositions predict trio compositions over evolutionary timescales.** It has been previously established that two-species

competition experiments could often predict the composition of multispecies communities[13,14]. While these experiments suggest that most ecological outcomes could be explained by pairwise interactions, it is still unknown whether similar predictability holds at evolutionary timescales. In order to assess predictability at evolutionary timescales, we employed an approach that was previously used to predict the composition of three-species communities from pairwise data at ecological timescales. In this approach, the fraction of each species in a trio is predicted to be the weighted geometric mean of its fraction when cocultured with each one of the species separately (Fig. 4A, "Methods").

Compositional changes that occur when species evolve in pairs are predictive of trio coevolution. In order to first evaluate whether incorporating data from evolutionary timescales is required, we used the fractions of pairs at generation ~70 to predict the trios' composition throughout the experiment. These predicted the trio compositions at the first ~70 generations with a prediction accuracy of 0.84 (similar to 0.86 that was found by Abreu et al.[14], Fig. 4B, Supplementary Fig. 9), significantly better than the uninformed guess (0.73 accuracy, "Methods", Mann–Whitney $p$ value $= 3 * 10^{-5}$). However, as anticipated by the fact that the composition changes significantly during these timescales (Fig. 2C), the accuracy of these predictions deteriorated with time; predictions of three-species community compositions after ~400 generations, based on pair fractions at ~70 generations were not significantly better than an uninformed guess (0.76 accuracy, Fig. 4B, Supplementary Fig. 9, Mann–Whitney $p$ value $= 0.1$). In contrast, predictions based on the pair fractions from the same time point as the trio were significantly more accurate than the uninformed guess throughout the whole experiment, and only slightly deteriorated with time

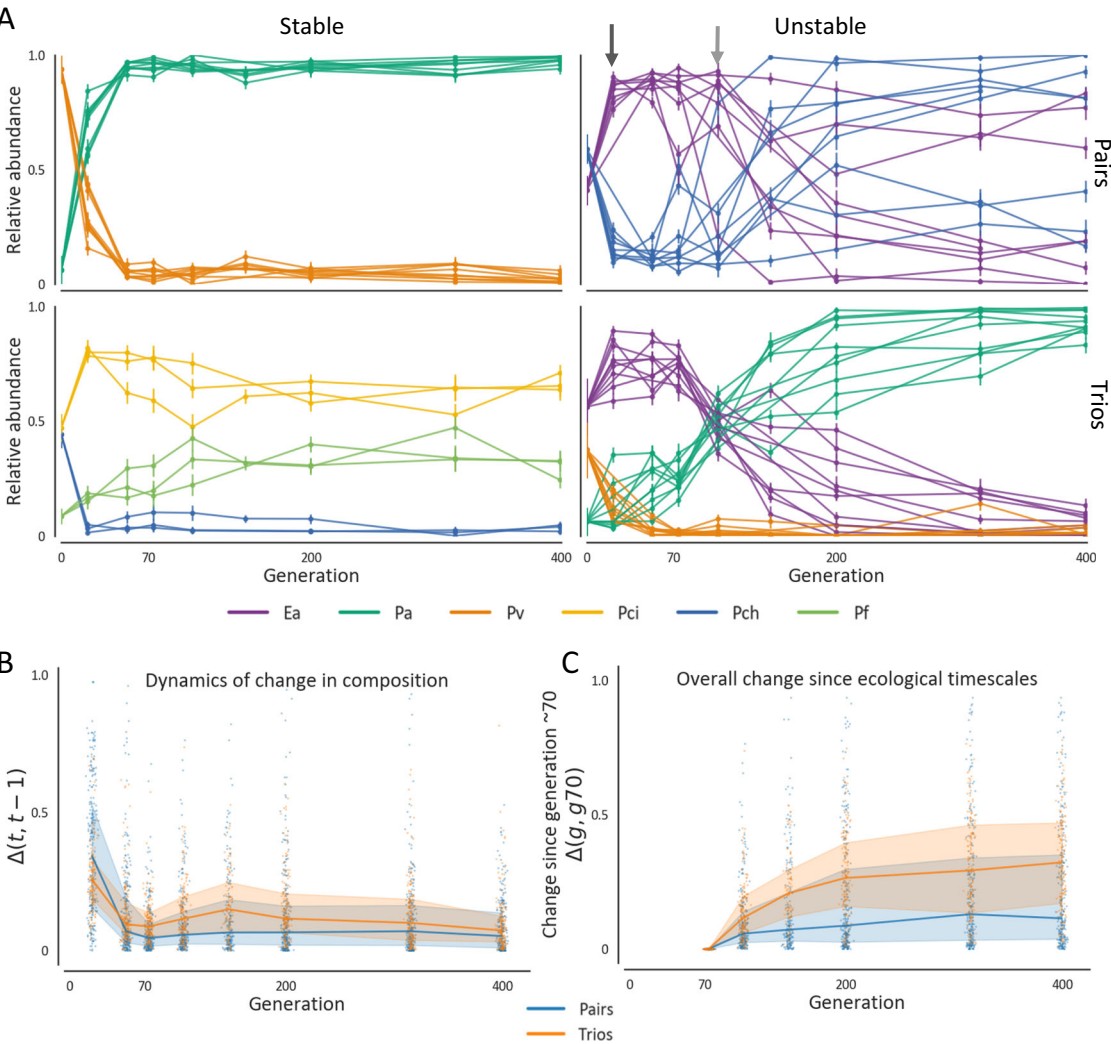

**Fig. 2 Most communities change significantly during the hundreds of generations following ecological equilibrium. A** Trajectories of the community composition of all replicates of four randomly picked communities representing different levels of stability and repeatability. Colors denote the different species in the community, and different lines are different replicates. Two arrows on the upper-right panel denote the time when most replicates have likely reached an ecological equilibrium (black arrow), and when this equilibrium was disrupted in most replicates (gray arrow). Error bars represent the standard deviation of the posterior beta distribution of the fractions, based on colony counts in each replicate, Error bars were calculated as $\sigma = \frac{\sqrt{p(1-p)}}{n+1}$, where p is the observed species fraction (colored dot) and n is the total number of colonies counted for a given replicate. For all cases $n > 15$. **B** Change in community composition across all communities quantified as the Euclidean distance between the composition of each replicate at two subsequent time points normalized to the maximal distance between two communities composed of n species ($\sqrt{n}$), denoted as $\Delta(t, t-1)$. **C** Change in community composition at evolutionary timescales measured as the Euclidean distance between the composition of each replicate in each time point and its composition at generation ~70 ($\Delta(g, g_{70})$). Generation ~70 is used here as the starting point of the evolutionary timescale since changes in most communities are less rapid after these timescales. For both **B** and **C**, blue and orange lines denote the median and shaded areas denote the interquartile range across all pairs and trios, respectively. Source data are provided as a Source Data file.

(0.83 accuracy at generation ~400, Fig. 4B, Supplementary Fig. 9, Mann–Whitney $p$ value = 0.004).

Furthermore, in 82% of trios the species whose abundance increased by the largest factor since ecological timescales could be accurately predicted based on the species' abundance increase in pairs (binomial test with p=1/3 $p$ value= $4 * 10^{-6}$, for 22 trios, "Methods"). These results demonstrate that the previously established assembly rule could be adjusted for predicting the composition of communities on evolutionary timescales, and suggests that high-order interactions are not major determinants of community composition during coevolution in our system.

**Growth of individually-evolved species does not predict community composition.** While pairwise competitions predict the

composition of multispecies communities, measuring the pairwise compositions of multiple pairs over evolutionary timescales is laborious and in many cases infeasible. Monoculture data, such as species' growth rates and carrying capacities, can be acquired more readily and was previously shown to correlate with the species' competitive ability[13–15]. Therefore, we next tested whether such monoculture data may provide a more accessible alternative for predicting the structure of multispecies communities.

We found that carrying capacities and growth rates could not accurately predict species' fractions or the identity of the dominant species in evolved coculture (Figs. 4, 5, Supplementary Figs. 11, 12, 13, 14). The accuracy of predicting species fractions based on the carrying capacities of individually-evolved species

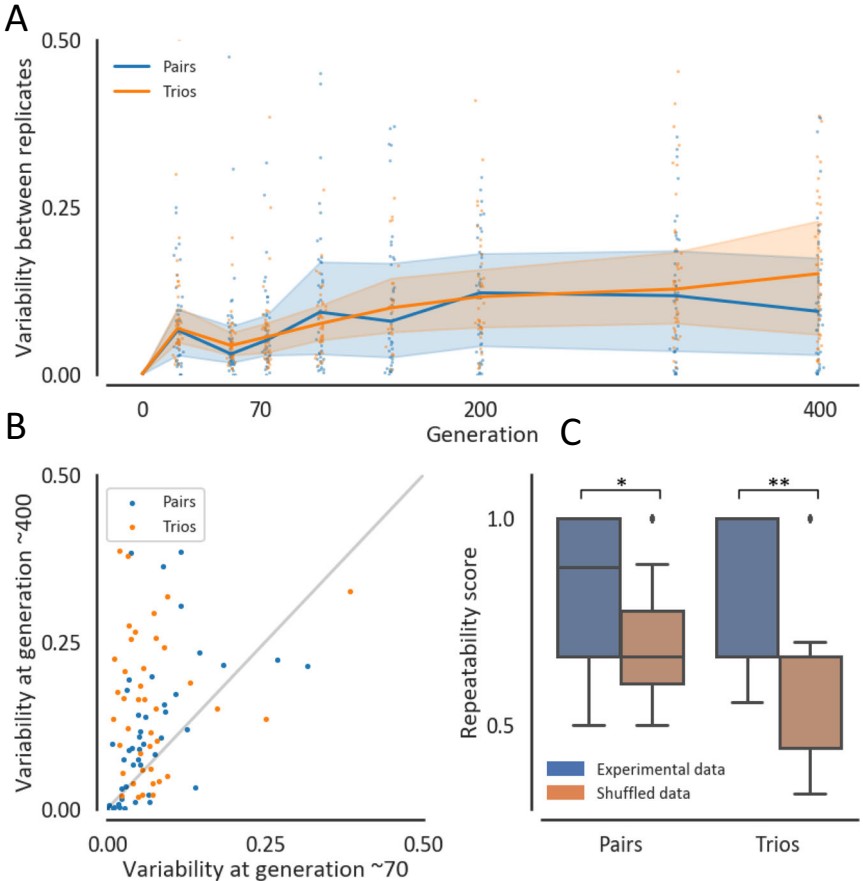

**Fig. 3 Variability between replicate communities increases during ~400 generations, yet remains significantly non-random. A** Variability in community composition between replicate communities is quantified by the mean Euclidean distance of each replicate from the medoid replicate normalized to the maximal distance between two communities composed of n species ($\sqrt{n}$). Blue and orange lines and shaded-areas are the medians and interquartile ranges across all pairs and trios, respectively. Dots denote the variability of specific communities across replicates. **B** Variability at generation ~400 against ~70, measured as the mean distance from medoid normalized to the maximal distance between two communities composed of n species ($\sqrt{n}$). Each dot represents a pair (blue) or a trio (orange). **C** Distribution of repeatability scores of the experimental data (38 pairs, 31 trios) and a random null model. The repeatability score is the frequency of replicates in which the same species increased its abundance by the biggest factor between generation ~70 and ~400. Communities that had missing replicates and therefore had less than 3 replicates for either generation ~70 or ~400, were removed from this analysis. The brown boxes represent the distribution of 2000 iterations of a shuffled model, where the values of changes in relative abundances between generation ~70 and ~400 of all species are pooled and are subsequently randomly assigned (for pairs and trios separately) to any species in any community in the dataset. Boxes indicate the quartiles and whiskers are expanded to include values no further than 1.5X interquartile range. *p value = 0.05, **p value <0.005. P values indicate the frequency of iterations were the shuffled data's mean repeatability score was at least as high as the experimental data's mean. Source data are provided as a Source Data file.

was comparable to that of an uninformed guess throughout the evolution experiment (Fig. 4B). The identity of the dominant species in two-species cocultures could be predicted with some accuracy at ecological timescales by the ancestral strains' carrying capacities and growth rates (Fig. 5A, Supplementary Figs. 11A, D, 14A, B, "Methods"). However, after evolving for ~400 generations, the identity of the dominant species could not be accurately predicted by the ancestral strains' growth ability, nor by the growth ability of strains that were evolved in monoculture (Fig. 5B, Supplementary Figs. 11B, C, E, F, 5, "Methods"). The deterioration in the predictive ability of the ancestral strain's growth ability after coevolution is consistent with the fact that almost all species increased in growth during ~400 generations of evolution (Supplementary Figs. 15, 5). The fact that the growth abilities of both individually-evolved strains and coevolved strains re-isolated from 21 pairs (Supplementary Fig. 13) also failed to predict pair compositions suggest that the evolved compositions are less determined by differences in growth capabilities than the ancestral compositions.

## Discussion

We observed that during ~400 generations of coevolution communities diverged from their ancestral compositions. However, it is not clear whether community compositions would continue to change at longer timescales of thousands of generations. Long-term evolutionary experiments in *E. coli* showed that following an initial period of rapid adaptation, evolutionary changes can continue for tens-of-thousands of generations at a lower rate[32]. In our experiment, changes in community composition slowed down slightly after ~200 generations (Figs. 2B, C, 3A), and it remains to be seen whether and on what timescales communities will reach evolutionary stable compositions.

While we found that community composition changed significantly in a repeatable manner in our experiment, there was considerable variability in stability and repeatability across the different communities (Figs. 2, 3, Supplementary Figs. 1, 2). It has been suggested that the strength and sign of interactions in a community could have a major influence on the evolutionary stability of the community[20,38]. Other factors, including the shape

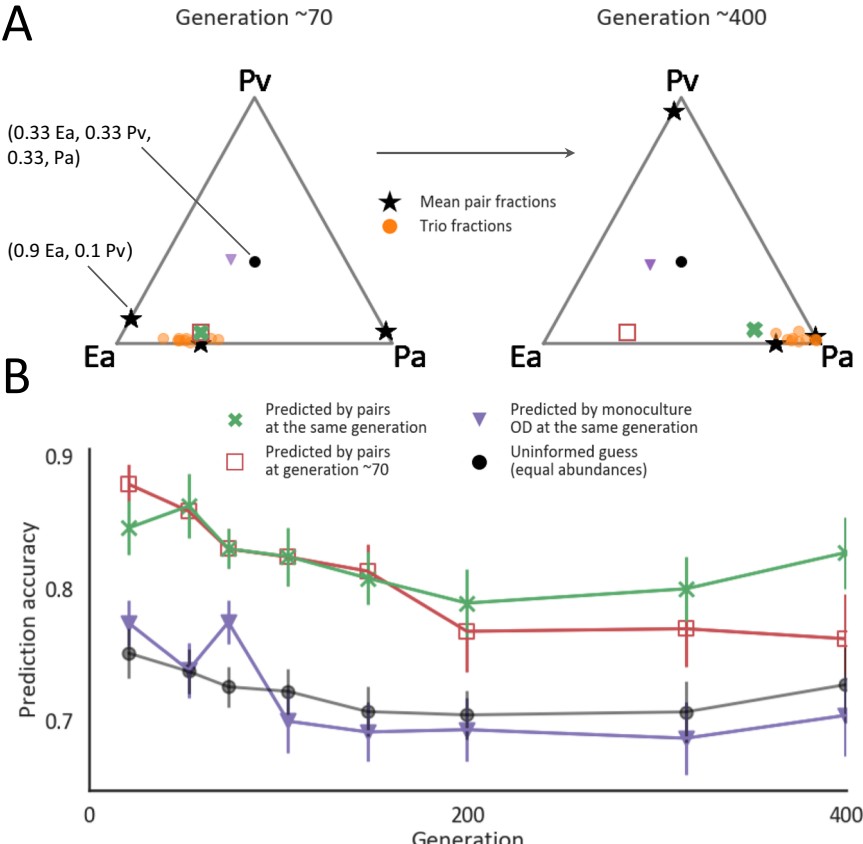

**Fig. 4 Changes in composition that occur in three-species communities are well predicted by those that occur in two-species communities. A** An example predicting the composition of one community at ecological and evolutionary timescales. The trio and the pairwise compositions of the three-species community Ea-Pa-Pv at generation ~70 and ~400. Each triangle is a simplex denoting the fractions of the three species, where each node is a specific species. The black stars on the edges denote the mean fractions in pairwise competition, the orange dots are the compositions of each of the trio replicates. The green cross is the predicted trio composition by pairs at the same generation and the red square, purple triangle, and black dot are predictions made by pairs at generation ~70, species mean carrying capacities at the same generation, and the best-uninformed guess (which means that the species fractions would all be 1/3) respectively. **B** The accuracy of prediction as $1 - \frac{\Delta(\text{prediction,observation})}{\sqrt{n}}$, where $\Delta$ (Prediction, Observation) is the Euclidean distance between the prediction and the observation and $\sqrt{n}$ is the largest possible distance between two communities with the same n species (here $n = 3$). Colored markers indicate the mean prediction accuracies of the predictions exemplified in (**A**), error bars indicate standard errors of 23 trios, and connecting lines help to visualize but do not have a biological meaning. Only trios composed of pairs whose evolutionary trajectories were measured were included in the analysis. Source data are provided as a Source Data file.

of the fitness landscape, species' evolvability, and their metabolic similarity could also affect the evolutionary trajectory of the community. Further work is needed in order to understand what makes certain communities evolutionary stable, and what makes some communities more evolutionary repeatable than others.

We have focused on the evolutionary dynamics and predictability of community composition, rather than community-level properties or traits of individual species within communities. While species abundances tend to change in a repeatable and predictable manner during coevolution, it is not clear whether the same is true for the underlying phenotypic and genotypic changes. For example, different mutations or changes in resource utilization may lead to similar changes in community composition. In contrast, the evolution of community-level properties, such as productivity, may be even more repeatable and predictable, as they have been shown to be less variable than species composition in natural communities[4,39].

Adaptation to the abiotic experimental conditions, such as media composition and growth-dilution cycles, may have played a significant role in driving changes in community composition. Recent experiments found that similar mutations were selected for when a pair of species evolved separately and in

coculture[40–43]. Furthermore, the fact that similar changes in community composition occurred in pairs and in trios may indicate that adaptation was not very sensitive to the specific biotic context and may have been driven by the abiotic conditions. However, the growth ability of individual species was only predictive of community composition on ecological timescales (Fig. 5, Supplementary Fig. 13). This suggests that when species coevolve, more complex interactions than competition for the supplied resources play a larger role in determining community composition. For example, species may adapt to consume or to better tolerate their partners' secretions. Elucidating the selective forces that act on species within communities remains an outstanding challenge[20].

Evolution in natural communities may differ significantly from our experiments, which involved highly simplified communities in stable laboratory conditions. In particular, our experiments were conducted in a well-mixed environment, thus reducing the possibility for spatial structure, which has been suggested to play an important role in the evolution of species interactions[44–47]. In addition, higher-order interactions may play a more significant role in shaping community composition in more species-rich communities, making predictions based on pairwise coevolution

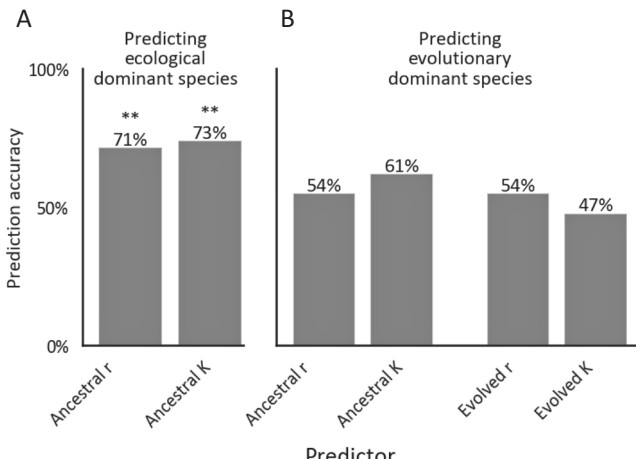

**Fig. 5 Growth of strains that evolved in monoculture does not predict the dominant species in pairs.** The accuracy of the prediction that the species with the higher growth rate (r)/carrying capacity (K) is more abundant in a pairwise competition at generation ~70 (**A**) and ~400 (**B**). Growth rate is measured as the time to a threshold of OD = 0.08 ("Methods"), and thus includes strains' lag time. An asterisks above bars indicate the probability to get a given level of accuracy by chance (one-sided binomial test, $n = 42$) ** $= <0.001$, for bars with no sign the probability is greater than 0.05. $P$ values (from left to right): 0.003, 0.001, 0.32, 0.08, 0.32, 0.67. One species' growth (H77) was not measured and therefore the two communities that included it were removed from this analysis, and accuracies were evaluated from the data of 42 pairs. Source data are provided as a Source Data file.

more challenging[13]. Furthermore, our assembly rule may be less predictive of communities involving more than one trophic-level, as was recently observed in an evolutionary experiment involving bacteria, a protist, and a phage[48].

In summary, microbial communities have the potential to be harnessed for numerous applications as plant growth promoters, bioremediation agents, biomanufacturing agents, or probiotic therapeutics. However, even in the absence of external perturbations, communities could change significantly over time and lose their desired functions. Our data highlights the need for knowledge of the evolutionary dynamics for a sustainable design of microbial communities, as it underlies the prevalence and timescales at which compositional changes occur. However, our findings suggest that compositional changes could still be predicted in a bottom-up approach from pairwise interactions even after hundreds of generations of coevolution. It still remains to be determined how well do these predictions scale-up to more diverse communities, how would different environmental conditions such as fluctuations and spatial structure influence these results, and the timescales at which these predictions still hold.

## Methods

**Strains and media.** The set of 16 strains used in this experiment contains environmental isolates along with strains from the ATCC collection (Supplementary Table 1). The strains were chosen based on two criteria: a distinct colony morphology that would enable visual identification when plated on an NB agar plate; and ability to coexist for ~60 generations with at least two other strains in our collection.

All cultures were grown in M9 minimal salts media containing 1X M9 salts, 2 mM $MgSO_4$, 0.1 mM $CaCl_2$, 1X trace metal solution (Teknova), supplemented with 3 mM galacturonic acid (Sigma), 6.1 mM Serine (Sigma), and 9.1 mM sodium acetate as carbon sources, which correspond to 16.67 mM carbon atoms for each compound and 50 mM overall. We chose a combination of carbon sources representing three chemical groups—a carbohydrate, an amino acid, and a carboxylic acid—in order to promote the survival and coexistence of a diverse set of species. The media was prepared on the day of each transfer. A carbon source mixture was prepared ahead at 10X, and was kept in aliquots at 4 °C for up to four weeks.

**Evolution experiment.** Frozen stocks of individual species were streaked out on nutrient agar Petri plates and grown at 28 °C. After 48 h single colonies were picked and inoculated into 15 ml falcon tubes containing 3 ml nutrient broth (5 g/L peptone BD difco, BD Bioscience; 3 g/L yeast extract BD difco, BD Bioscience), and were grown overnight at 28 °C shaken at 250 rpm. Initial mixtures were prepared by diluting each species separately to an OD of $10^{-2}$ and mixing the normalized cultures at equal volumes. OD measurements were done using a Epoch2 microplate reader (BioTek) and were recorded using the Gen5 v3.09 software (BioTek). After mixing, the cocultures were aliquoted to replicates and further diluted to a final OD of $10^{-4}$, at which the evolutionary experiment was initialized. The number of replicates for each community varied between 3 and 18 (Supplementary Data 1).

Communities were grown in 96-well plates containing 200 μl M9 at 28°C and were shaken at 900 rpm. Every 48 h cultures were diluted by a factor of 1500 into fresh M9 media, and $OD_{600}$ was measured. For this dilution factor, each cycle corresponds to ~10.5 generations. As 1 $OD_{600}$ ~ of $10^9$ C.F.U/ml, and communities reached ~ 0.5 $OD_{600}$ and were grown in 200 μl and was diluted by 1500, ~$10^5$ cells were transferred each dilution. To avoid cross contaminations, cultures were grown in a checkerboard formation, meaning that each community was surrounded by wells containing media but no bacteria.

At transfers 0, 2, 5, 7, 10, 14, 19, 30, and 38 community composition was measured by plating on nutrient agar plates (5 g/L peptone BD difco, BD Bioscience; 3 g/L yeast extract BD difco, BD Bioscience, 15 g/L agar Bacto, BD Bioscience) and counting colonies. For that, the cultures were diluted to an OD of $2.4 * 10^{-8} - 1 * 10^{-8}$ and 100 μl of the diluted culture was plated on NB plates and spread using glass beads. Plates were incubated at 28 °C for 48 h and colonies were counted manually. The distribution of the number of colonies counted at each plate to infer community composition is found in Supplementary Fig. 11.

We chose the communities based on a preliminary experiment that was conducted by the same protocol for six transfers. In this experiment, 114 of 171 possible pairs of a set of 19 strains (3 strains were not included in the evolution experiment) were cocultured. Pairs that had coexisted for the duration of this experiment, and were confidently distinguishable by colony morphology, and trios that are composed of these pairs, were used for the coevolutionary experiment. We started the evolutionary experiment with 51 pairs and 51 trios, and removed communities that did not coexist for the first ~70 from the final analysis. If a replicate was suspected to be contaminated it was also excluded from further analysis.

**Ecological experiments.** We supplemented the data of the evolutionary experiment with two ecological competition experiments with the same experimental condition. In order to assess whether communities typically reach an ecological equilibrium within ~50–70 generations (Supplementary Fig. 3), we cultured eight of the pairs that were used in the evolutionary experiment. This experiment was initiated in the same way as the evolutionary experiment, only that after the species' starters were normalized they were inoculated at the varying initial fractions - 9:1, 5:5, 1:9. Because the normalization depended on optical density, there is a variation in the actual initial fractions between different pairs. Community composition was then measured on six transfers during this experiment: 0, 1, 2, 4, 5, and 6.

In order to assess whether changes in composition are due to heritable changes in species' phenotypes, we used strains that were re-isolated from 31 evolved pairs, and 13 pairs of ancestral strains (Supplementary Figs. 6, 7). Strains were replicated from glycerol stocks into the experimental media and grown for 24 h. The starters were normalized to initiate the competition assay at OD = $10^{-4}$ in fresh M9 media. Species were mixed at equal volume and were propagated for five cycles. community composition was measured at initial conditions, and at the end of the final cycle (5).

**Quantification of repeatability.** In order to quantify the qualitative repeatability of different replicate communities we first identified which species was the maximally increasing member at each replicate, that is, which species had increased its abundance by the largest factor between generation 70 and 400. Then, we quantified the frequency of the replicates that had the same maximally increasing member for each community. This measure always produces a value between 1 and $1/n$ where $n$ is the number of species in the community. We checked the distribution of the repeatability scores against the null hypothesis that the factor by which a species' abundance increases during evolution is independent of the species or the community. For this, we shuffled the factor of change in relative abundance across all samples, for pairs and trios separately, and quantified the new repeatability scores of the shuffled data. Data of the null hypothesis were generated over 2000 times, and the $p$ value was given by the probability to get a mean equal or above the real data mean.

We used the average Euclidean distance of replicates from the median replicate in order to quantify the variability between replicate communities. In order to check whether the distribution of variabilities is similar to what can be expected of random communities, in which each species in the community is just as likely to have any relative abundance, we replaced the real fractions with fractions drawn from a uniform Dirichlet distribution with $\underline{\boldsymbol{\alpha} = 1}$. We then checked the statistical difference between the two distributions using one-sided Mann–Whitney U test.

**Trio composition predictions**. We used the formerly established method for predicting the composition of trios from the composition of pairs that was developed by Abreu et al.[14] In this approach the fraction of a species when grown in a multispecies community is predicted as the weighted geometric mean of the fraction of the species in all pairwise cultures. The accuracy of the predictions was measured as the Euclidean distance between the prediction and the mean composition of the observed trio, normalized to the largest possible distance between each two communities, $\sqrt{n}$, where n is the number of species.

We used the factors by which species increased their abundance during coevolution in pairs (between generations ~70 and ~400) to predict which species would increase by the largest factor in trios. The maximally increasing member in a given community was assigned to be the one that was the maximally increasing member in the most replicates of that community. If the same species was the maximally increasing member in both pairs it was a member of, then this species was predicted to be the maximally increasing member of the trio. If in every pair a different species was the maximally increasing member, then we predicted that the maximally increasing member of the trio would be the one with the highest mean increase. Only two trios had such transient topology, where in each pair a different species increases, thus we are unable to determine the general utility of the latter approach.

**Re-isolation**. Each ~50 generations all communities were frozen at −80 °C with 50% glycerol in a 96-deep well plate. In order to re-isolate strains, stocks were inoculated to a 96-well plate containing the experimental media using a 96-pin replicator, and grown for 24 h at 28 °C. After growth, cultures were diluted by a factor of $2.4 * 10^{-8}$ and 100 µl were spread on a nutrient agar plate using glass beads. Plates were kept at room temperature for at least two days and no longer than a week before re-isolations. 5-15 colonies of each strain were picked using a sterile toothpick, and pooled together into 200 µl M9. Re-isolated strains were incubated at 28 °C and shaken at 900 rpm for 24 h and kept in 50% glycerol stock at −80 °C until further use.

**Growth rates and carrying capacities of individually evolved strains**. Re-isolated strains were replicated from glycerol stocks into the experimental media and grown for 24 h. The starters were normalized to initiate the growth assay at $OD = 10^{-4}$ in fresh M9 media. The optical density was measured in two automated plate readers simultaneously, Epoch2 microplate reader (BioTek) and Synergy microplate reader (BioTek), and was recorded using Gen5 v3.09 software (BioTek). Plates were incubated at 28 °C with a 1 °C gradient to avoid condensation on the lid, and were shaken at 250 cpm. OD was measured every 10 min. Each strain was measured in four technical replicates, evenly distributed between the two plates, and 2–3 evolutionary replicates were measured for each species (replicates that evolved separately for the duration of the experiment). Growth rates were quantified as the number of divisions it takes a strain to grow from the initial OD of $10^{-4}$ to an OD of $8 * 10^{-2}$ ($\log_2 \frac{0.08}{10^{-4}}$) divided by the time it took the strain to reach this OD. This measure gives the average doubling time during the initial growth and also accounts for the lag times of the strain. The growth rates of evolutionary replicates were averaged after averaging technical replicates.

Carrying capacity was defined as the OD a monoculture reached at the end of each growth cycle of the evolutionary experiment averaged across replicates. These measurements were done in an Epoch2 microplate reader (BioTek). In order to reduce noise, the trajectories of OD measurements were smoothed for each well using moving mean with an averaging window of three.

**Carrying capacities of coevolved strains**. Re-isolated strains were replicated from glycerol stocks into the experimental media and grown for 48 h in M9 media at 28 °C. Cultures were then diluted by 1500 into 3 technical replicates in fresh M9-media, and were given another 48 h to reach carrying capacity. The strains used in this experiment were isolated from 1-3 evolutionary replicates (Supplementary Data 2).

**Reporting summary**. Further information on research design is available in the Nature Research Reporting Summary linked to this article.

## Data availability
The full dataset used in this paper is available at github.com/nittaym/Evo_assembly_rules, https://doi.org/10.5281/zenodo.4704257. Source data are provided with this paper.

## Code availability
The Python 3 code used for the analysis is available at github.com/nittaym / Evo_assembly_rules, https://doi.org/10.5281/zenodo.4704257.

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

## Acknowledgements

We thank Daniel Rodríguez Amor for constructive comments on the manuscript, and Nadav Kashtan, Alfonso Pérez Escudero, and members of the Friedman lab for helpful discussions. This research was supported by the Israel Science Foundation (grant No. 81/097).

## Author contributions

N.M. and J.F. designed the study and wrote the manuscript, N.M., N.T., and Y.S. performed the experiments, and N.M performed the analysis.

## Competing interests

The authors declare no competing interests.
