## [Peer Review File · Nature Communications]

Reviewers' Comments:

Reviewer #1:

Remarks to the Author:

The authors address a highly important question in microbial ecology and evolution: how repeatable (and therefore predictable) evolution is within communities. To this end, they performed a large number of experimental community evolution experiments using defined synthetic consortia with two or three members. The authors find that evolution in replicate 3-member communities leads to similar outcomes, which are predictable from the outcome of pairwise evolution. In addition, the paper addresses what I believe is a major caveat (and seldom studied open question) of synthetic ecology: the fact that evolution is believed to be rapid in bacteria, and therefore it may fundamentally alter the consortia one creates.

This is a very interesting and timely paper and I have thoroughly enjoyed reading it. The question the authors ask is both extremely important and understudied. The experiments are well designed and the results are clean. The interpretation is convincing. The paper is also clearly written and accessible to a broad audience, and the figures are clear and helpful. I am rather enthusiastic about this paper and supportive of its publication and I only offer some minor suggestions to improve readability and enhance the message of the paper (adopting them is optional)

1) Because the emphasis is on repeatability/reproducibility, I think the authors should highlight the number of replicates of each community in the abstract (e.g. "XXX parallel evolution experiments were carried out for each of 87 different two and three species communities"). Likewise, the authors may want to mention the number of replicates in lines 57-59 of the main text.

2) In Line 71, it would be helpful if the authors said a bit more of the minimal medium used (i.e. limiting factors), the volume, and the dilution factor used. One can see that in the Methods but it would facilitate reading if the info was given here in the main text.

3) In the caption of Fig. 2A (Lines 95-100), what do the error bars represent? They seem small, in my experience with plating, errors tend to be rather large owing to sampling...

4) Also with regard to Fig. 2A (trios): Is it clear that the changes in composition observed after the 70th generation are due to evolution? Couldn't they be a slower ecological timescale in the trio, which becomes more apparent when Pv drops below certain level (also occurring around generation 70) originating a de facto transition from 3 to ~2 species dynamics (acknowledging that Pv is still there, but now at much lower abundance, so its effect on the other two is diminished)? I would expect that as communities become more diverse, the ecological relaxation dynamics will be longer, so the equilibration time for pairs and trios may not be the same

I saw that this matter has been addressed in Fig. S5. I really loved this experiment, and I found this figure rather important, and I would encourage the authors to move it to the main text, space permitting. It would have been nice to show here also an example or two where the authors mixed ancestral strains or their derived strains and compare their short-term dynamics, overlaying on the same plot the ancestral "equilibrium"...

To summarize: I think it would be helpful to show a few representative examples of the same pair and trio inoculated with ancestral or derived strains, and show that they do converge to different equilibria over short timescales (which would show that the effect observed is indeed evolutionary rather than slow ecological dynamics) together with the "summary" panel they now have in Fig. S5. I think it would be nice to have this in the main text, space permitting, but if the authors prefer to keep it in the supplement I would advise adding a few representative examples to S5.

5) I also found Fig. S6 more informative than Fig. 3. Of course this is the authors choice, but although

the random control is a sanity check, I find it could be in the supplement whereas visualizing the pairs and trios separately is more interesting (or it was to me, for whatever is worth!). For instance, you can see in panel B that controlling for similar variability at gen 70, trios are more variable than pairs at gen 400 (unless I am misinterpreting this plot...).

6) I also would mark generation 70 in panel A (both in Fig. 3 and Fig. S6)

7) What are the N for each bar in Fig. 5?

8) In the discussion the authors may want to speculate about what may be causing the finding that having a high r/K is a much better predictor for ecological dominance in experiments with ancestral strains than with derived strains? For instance, is it possible that co-evolution is making species adapting to each other's secretions?

In sum, I very much enjoyed this very interesting paper and I am positive about its publication.

Alvaro Sanchez
Department of Ecology & Evolutionary Biology
Yale University

Reviewer #2:

Remarks to the Author:

This review is prepared by Wenying Shou

Meroz et al. examined 87 two and three-species bacterial communities (from combinations of 16 soil bacterial species that were found to coexist for at least 60 generations) for 400 generations in a well-mixed environment. Community composition initially changed due to ecology, and later changed due to evolution. During evolution, variability in species composition among replicate community increased over time. However, replicate communities remained similar enough to each other to allow predictions to be made from evolved pairwise communities about evolved trio communities. Interestingly, growth rate and carrying capacity of singly evolved species could not accurately predict the identity of dominant species in evolved coculture, suggesting that coevolution may have occurred.

I think that this work nicely continues the authors' earlier work on community assembly rules on ancestral communities. I do not think that new experiments are needed. However, the writing can be significantly improved. The main text is too succinct, and the reader is forced to look for information in other places of the text. Examples are given below:

1. Figure 1 legend: "All 16 species were evolved in monocultures" – for how long? The assembly of communities happened after monoculture evolution, is that correct? Notes added later: After I got to Fig 4, I then realized that monoculture evolution was likely carried out in parallel, and when I got to Fig 5, I was confident about this. This is not ideal.

2. Figure 2A legend: After staring at the graph for a while, one then realizes that perhaps there is no systematic relationship between the top and the bottom panels (besides the possible involvement of the same species). Because of the label of "pairs" and "trio", one naturally wonders about the three pairs per trio relationship. Rather, these figures are more or less randomly picked. If so, please state "randomly picked from representative classes".

3. Figure 3 legend: Fully shuffled model: how often do you shuffle? Do you preserve some temporal dependence between adjacent time points during shuffling?

4. Fig 4A: red triangle not easily visible

Minor:

Line 28-29: This statement is correct. However, I want to point out that quantitative mechanistic modelling is possible, although this is tremendously challenging even for a highly simplified community (Hart et al., PLoS Biology 2019). I feel that giving a comprehensive intro is important.

Line 256: The single citation can be expanded to suggest that the phenomenon is not esoteric. For example, we have found multiple examples of the same mutations being selected for in monocultures and in cocultures (Green & Sonal et al., PLoS Biology, 2020, <https://journals.plos.org/plosbiology/article/comments?id=10.1371/journal.pbio.3000757>; Hart et al., eLife 2021, <https://elifesciences.org/articles/57838>).

Reviewer #3:

Remarks to the Author:

This manuscript investigates evolutionary dynamics in microbial communities. The authors track species abundance as 16 bacterial strains are evolved for ~400 generations in monocultures, and 2- and 3-species co-cultures. Community compositions change rapidly for ~70 generations and then become more stable on average, though there is substantial variation in this pattern between communities. Replicate communities become less similar in composition over time, however even at 400 generations replicate communities are more similar than expected by chance. Additionally, within replicates there was repetition of which species increased the most in abundance over the course of the experiment. Species abundance in evolving 2-species communities predicted species abundance in evolving 3-species communities better than a null uninformed model. Finally, the growth rate and carrying capacity of strains evolved in monoculture provided little information on the relative abundance of strains in evolved communities.

This work addresses exciting questions about the degree to which evolution in a community context is repeatable and predictable. The experiment represents a tour de force with 87 communities and 16 monocultures all experimentally evolved with at least 3-fold replication. The manuscript provides interesting analyses demonstrating that even in a community context there is a degree of both repeatability and predictability to evolution.

With large data sets such as this there are many different analyses that could be done and no doubt the authors will continue to mine this data for many additional papers. There are however, a few analyses that I think would significantly enhance the current work. First I would be interested to see data on the abundance change over the course of evolution for each species in monoculture, 2- and 3-species co-cultures. Much of this data is already present but typically presented as ratios, and not broken down by species. How do the distributions of change in co-cultures compare between species? Are there tight correlations in change across treatments for some species? These are clearly data that the authors have in hand so this should represent little additional work.

I would also be highly interested in the phenotypes of evolved isolates. How do growth rate and efficiency in monoculture change for populations evolved in monoculture and co-culture? This is relevant for example for thinking about the predictive value of r and K in the final section. These parameters for strains evolved in monoculture, do not predict the composition of evolved communities. However, it is unclear whether this lack of prediction is because populations evolve differently in monoculture and co-culture, or because growth rate and carrying capacity are poor predictors of community composition in the evolved communities. It may not be feasible to determine growth rate and efficiency for all evolved isolates in monoculture, but even a representative subset would be informative.

In the methods there are 2 null distributions described for determining repeatability, however it seems

that only one comparison is presented in the paper. The supplement also has a comparison against a null distribution, but it appears it is the same comparison just with the data broken down into 2 and 3 species scenarios. Am I missing the second distribution?

Line 214 – I would encourage the authors to highlight that their measure of growth rate includes lag time. This is clearly explained in the methods, but I think it would help readers interpret results to mention it when the results are being discussed.

Typo in line 295.

We thank the reviewers for their positive assessment of our manuscript and for constructive comments that have helped us improve this work. Below we provide a point-by-point reply to the Reviewers' comments.

REVIEWER COMMENTS

Reviewer #1 (Remarks to the Author):

The authors address a highly important question in microbial ecology and evolution: how repeatable (and therefore predictable) evolution is within communities. To this end, they performed a large number of experimental community evolution experiments using defined synthetic consortia with two or three members. The authors find that evolution in replicate 3-member communities leads to similar outcomes, which are predictable from the outcome of pairwise evolution. In addition, the paper addresses what I believe is a major caveat (and seldom studied open question) of synthetic ecology: the fact that evolution is believed to be rapid in bacteria, and therefore it may fundamentally alter the consortia one creates.

This is a very interesting and timely paper and I have thoroughly enjoyed reading it. The question the authors ask is both extremely important and understudied. The experiments are well designed and the results are clean. The interpretation is convincing. The paper is also clearly written and accessible to a broad audience, and the figures are clear and helpful. I am rather enthusiastic about this paper and supportive of its publication and I only offer some minor suggestions to improve readability and enhance the message of the paper (adopting them is optional)

1) Because the emphasis is on repeatability/reproducibility, I think the authors should highlight the number of replicates of each community in the abstract (e.g. "XXX parallel evolution experiments were carried out for each of 87 different two and three species communities"). Likewise, the authors may want to mention the number of replicates in lines 57-59 of the main text.

This is indeed important information to highlight. We added the number of replicates in the abstract and in the main text:

"we address this knowledge gap by tracking the composition of 87 two- and three-species bacterial communities, with 3-18 replicates each, for ~400 generations" (lines 13-14)

"we use experimental evolution of 87 two- and three-species bacterial communities, with 3-18 replicates each". (lines 58-59)

2) In Line 71, it would be helpful if the authors said a bit more of the minimal medium used (i.e. limiting factors), the volume, and the dilution factor used. One can see that in the Methods but it would facilitate reading if the info was given here in the main text.

We added this information:

“well-mixed microcosms containing M9 minimal-media supplemented with three carbon sources - galacturonic acid, acetate, and serine” (lines 70-71) ... “were cultured in 200µl minimal media and passaged through 38 cycles of 48h of growth and 1500-fold dilution” (lines 74-75)

3) In the caption of Fig. 2A (Lines 95-100), what do the error bars represent? They seem small, in my experience with plating, errors tend to be rather large owing to sampling...

We apologize for not properly explaining the meaning of these error bars. These error-bars are analytical estimates of the error in species' relative abundance in a single replicate, assuming random sampling from the community. They do not represent the error between repeated measurements or replicates, which indeed tend to be larger. Nonetheless, we feel that they are informative, as they give the reader a sense of the number of colonies that were counted and the certainty in the estimated relative abundances of each replicate. We have added more information regarding these error bars to the figure caption:

“Error bars represent the standard deviation of the posterior beta distribution of the fractions, based on colony counts in each replicate. Error bars were calculated as $\sigma = \frac{\sqrt{p(1-p)}}{n+1}$, where p is an observed species fraction and n is the total number of colonies counted for a given replicate”. (Lines 103-105 in the main text, and lines 4-7 and 10-13 in the supplementary file)

4) Also with regard to Fig. 2A (trios): Is it clear that the changes in composition observed after the 70'th generation are due to evolution? Couldn't they be a slower ecological timescale in the trio, which becomes more apparent when P_v drops below certain level (also occurring around generation 70) originating a de facto transition from 3 to ~2 species dynamics (acknowledging that P_v is still there, but now at much lower abundance, so its effect on the other two is diminished)? I would expect that as communities become more diverse, the ecological relaxation dynamics will be longer, so the equilibration time for pairs and trios may not be the same

I saw that this matter has been addressed in Fig. S5. I really loved this experiment, and I found this figure rather important, and I would encourage the authors to move it to the main text, space permitting. It would have been nice to show here also an example or two where the authors mixed ancestral strains or their derived strains and compare their short-term dynamics, overlaying on the same plot the ancestral “equilibrium”...

To summarize: I think it would be helpful to show a few representative examples of the same pair and trio inoculated with ancestral or derived strains, and show that they do converge to different equilibria over short timescales (which would show that the effect observed is indeed evolutionary rather than slow ecological dynamics) together with the “summary” panel they now have in Fig. S5. I think it would be nice to have this in the main text, space permitting, but if the

authors prefer to keep it in the supplement I would advise adding a few representative examples to S5.

Thank you for this detailed comment, we agree that this is important information to include. We added a new supplementary figure (also included below) which shows the distribution of fractions after ~53 generations (5 transfers) for pairs of ancestral strains and pairs of strains reisolated from gen ~400 of the original evolutionary experiment alongside the fractions of those pairs measured at generation ~53 and ~400 of the long experiment.

Overall, after ~53 generations, the fractions of pairs of strains that have evolved in coculture are more similar to the fractions of these pairs at generation ~400 than at generation ~53 of our original evolution experiment. However, that is not always the case (e.g the pairs Ea+Pv and Ea+Pch). We are still investigating the cause of these intriguing results, but one possibility is that they result from the presence of within-population heterogeneity in the original evolution experiment that is lost in the new experiments conducted with single colonies isolated from these experiments. Long ecological dynamics in the original experiment seem unlikely, since there was considerable variability between replicates in that experiment. But clearly this requires further investigation.

Since these experiments take a considerable amount of effort, we limited them to a subset of the pairs and measured their fractions only after ~53 generations. We did not measure the composition of reisolated strains that have coevolved in trios. Therefore, we chose to keep this figure in the supplement. We hope that these data are sufficient to address the reviewers' concerns.

Figure S7. The composition reached by ancestral strains and coevolved strains after a ~53 generation experiment, and the composition of the same pairs at generations ~53 and ~400 of the long experiment. Distributions represent the fraction of one of the species. Coevolved strains are strains that were reisolated from coculture at generation ~400. The composition of coevolved strains was measured in two technical replicates and a varying number of evolutionary replicates.

5) I also found Fig. S6 more informative than Fig. 3. Of course this is the authors choice, but although the random control is a sanity check, I find it could be in the supplement whereas visualizing the pairs and trios separately is more interesting (or it was to me, for whatever is worth!). For instance, you can see in panel B that controlling for similar variability at gen 70, trios are more variable than pairs at gen 400 (unless I am misinterpreting this plot...).

We actually had a hard time deciding which of these would be in the main text and which would be in the supplementary, so we appreciate this comment that convinced us to switch between the two. We switched Fig. S6 with Fig. 3, and also Fig. S4 with Fig. 2B,C (which also shows the data broken down to pairs and trios, or not broken down)

6) I also would mark generation 70 in panel A (both in Fig. 3 and Fig. S6)

Great suggestion, we added it on both figures.

7) What are the N for each bar in Fig. 5?

We added the following clarification to the figure caption:

“accuracies were evaluated from the data of 42 pairs” (line 223)

8) In the discussion the authors may want to speculate about what may be causing the finding that having a high r/K is a much better predictor for ecological dominance in experiments with ancestral strains than with derived strains? For instance, is it possible that co-evolution is making species adapting to each other's secretions?

Thank you for this comment and suggested mechanism. We added this hypothesis to our discussion:

“However, the growth ability of individual species was only predictive of community composition on ecological timescales (Fig. 5, S13). This suggests that when species coevolve, more complex interactions than competition for the supplied resources plays a larger role in determining community composition. For example, species may adapt to consume or to better tolerate their partners' secretions”. (lines 269 - 274)

In sum, I very much enjoyed this very interesting paper and I am positive about its publication.

Alvaro Sanchez
Department of Ecology & Evolutionary Biology
Yale University

Reviewer #2 (Remarks to the Author):

This review is prepared by Wenying Shou

Meroz et al. examined 87 two and three-species bacterial communities (from combinations of 16 soil bacterial species that were found to coexist for at least 60 generations) for 400 generations in a well-mixed environment. Community composition initially changed due to ecology, and later changed due to evolution. During evolution, variability in species composition among replicate community increased over time. However, replicate communities remained similar enough to each other to allow predictions to be made from evolved pairwise communities about evolved trio communities. Interestingly, growth rate and carrying capacity of singly evolved species could not accurately predict the identity of dominant species in evolved coculture, suggesting that coevolution may have occurred.

I think that this work nicely continues the authors' earlier work on community assembly rules on ancestral communities. I do not think that new experiments are needed. However, the writing can be significantly improved. The main text is too succinct, and the reader is forced to look for information in other places of the text. Examples are given below:

Following this comment and comments from other reviewers we added more information about key aspects of the experiments and analysis in the main text. We hope these additions improve the readability of the manuscript.

1. Figure 1 legend: "All 16 species were evolved in monocultures" – for how long? The assembly of communities happened after monoculture evolution, is that correct? Notes added later: After I got to Fig 4, I then realized that monoculture evolution was likely carried out in parallel, and when I got to Fig 5, I was confident about this. This is not ideal.

We have added a clarification to the caption of Fig 1:

"Subsets of the 16 species were used to assemble 44 pairs and 43 trios. All 16 species were evolved in monocultures in parallel to the assembled communities." (lines 84-85)

2. Figure 2A legend: After staring at the graph for a while, one then realizes that perhaps there is no systematic relationship between the top and the bottom panels (besides the possible involvement of the same species). Because of the label of "pairs" and "trio", one naturally wonders about the three pairs per trio relationship. Rather, these figures are more or less randomly picked. If so, please state "randomly picked from representative classes".

We thank the reviewer for bringing to our attention that this was not clear. We adopted the reviewer's suggestion and added this at the caption:

“Trajectories of the community composition of all replicates of four randomly picked communities representing different levels of stability and repeatability.” (line 99)

3. Figure 3 legend: Fully shuffled model: how often do you shuffle? Do you preserve some temporal dependence between adjacent time points during shuffling?

We apologize this wasn't sufficiently clear. For each community, our analysis only considers the relative abundances at generations ~70 and ~400. It does not account for the dynamics between these two time points, and therefore there is no temporal dependence between adjacent time points to consider. We added the following lines to the caption in order to add missing details and clarify this analysis:

“The brown boxes represent the distribution of 1500 iterations of a shuffled model, where the values of changes in relative abundances between generation ~70 and ~400 of all species are pooled and are subsequently randomly assigned (for pairs and trios separately) to any species in any community in the dataset (Methods)” (lines 161-163)

A more detailed description is also present in the methods section:

“In order to quantify the qualitative repeatability of different replicate communities we first identified which species was the maximally increasing member at each replicate, that is, which species had increased its abundance by the largest factor between generation 70 and 400. Then, we quantified the frequency of the replicates that had the same maximally increasing member for each community. This measure always produces a value between 1 and 1/n where n is the number of species in the community. We checked the distribution of the repeatability scores against the null hypothesis that the factor by which a species' abundance increases during evolution is independent of the species or the community. For this, we shuffled the factor of change in relative abundance across all samples, for pairs and trios separately, and quantified the new repeatability scores of the shuffled data.. Data of the null hypothesis were generated over 1500 times, and the p-value was given by the probability to get a mean equal or above the real data mean.” (lines 358-369)

4. Fig 4A: red triangle not easily visible

Yes, the red triangle is indeed hard to spot on the right ternary plot because it is equivalent to the green cross there. The predictions made by pairs at the same generation are equivalent to those made by the pairs at generation ~70 when in generation ~70. We decided to leave the triangle there, although it contains no additional information, only to help the reader understand what is meant by 'predicted by pairs at generation ~70'.

To try to make it more visible we switched the marks so that the 'predicted by pairs at generation ~70' is now an empty square (see below).

Figure 4: Changes in composition that occur in three-species communities are well predicted by those that occur in two-species communities. (A) An example predicting the composition of one community at ecological and evolutionary timescales. The trio and the pairwise compositions of the three-species community Ea-Pa-Pv at generation ~70 and ~400. Each triangle is a simplex denoting the fractions of the three species, where each node is a specific species. The black stars on the edges denote the mean fractions in pairwise competition, the orange dots are the compositions of each of the trio replicates. The green cross is the predicted trio composition by pairs at the same generation and the red square, purple triangle and black dot are a prediction made by pairs at generation ~70, prediction made by using species mean carrying capacities at the same generation, and the best-uninformed guess which means that the species fractions would all be $\frac{1}{3}$, respectively. (B) The accuracy of prediction as $1 - \frac{\Delta(\text{prediction, observation})}{\sqrt{n}}$, where $\Delta(\text{Prediction, Observation})$ is the Euclidean distance between the prediction and the observation and \sqrt{n} is the largest possible distance between two communities with the same n species (here $n=3$). Colored markers indicate the mean prediction accuracies of the predictions exemplified in A, error bars

indicate standard errors of 23 trios, and connecting lines help to visualize but do not have a biological meaning. Only trios composed of pairs whose evolutionary trajectories were measured were included in the analysis.

Minor:

Line 28-29: This statement is correct. However, I want to point out that quantitative mechanistic modelling is possible, although this is tremendously challenging even for a highly simplified community (Hart et al., PLoS Biology 2019). I feel that giving a comprehensive intro is important.

Thank you for pointing out this work. It is indeed very relevant to the point made in this paragraph and we have now cited it. Furthermore, we hope that this point is clear when we state that:

“Challenges in fitting such models...have recently led us to develop an alternative non-parametric predictive framework” (line 30-31)

Line 256: The single citation can be expanded to suggest that the phenomenon is not esoteric. For example, we have found multiple examples of the same mutations being selected for in monocultures and in cocultures (Green & Sonal et al., PLoS Biology, 2020, <https://journals.plos.org/plosbiology/article/comments?id=10.1371/journal.pbio.3000757>; Hart et al., eLife 2021, <https://elifesciences.org/articles/57838>).

We agree that showing that this phenomenon is widespread can contribute to the discussion, thanks for pointing out these relevant papers. We added these citations and changed the statement accordingly (line 266-267):

“Recent recent experiments found that similar mutations were selected for when a pair of species evolved separately and in coculture”

Reviewer #3 (Remarks to the Author):

This manuscript investigates evolutionary dynamics in microbial communities. The authors track species abundance as 16 bacterial strains are evolved for ~400 generations in monocultures, and 2- and 3-species co-cultures. Community compositions change rapidly for ~70 generations and then become more stable on average, though there is substantial variation in this pattern between communities. Replicate communities become less similar in composition over time, however even at 400 generations replicate communities are more similar than expected by chance. Additionally, within replicates there was repetition of which species increased the most in abundance over the course of the experiment. Species abundance in evolving 2-species communities predicted species abundance in evolving 3-species communities better than a null uninformed model. Finally, the growth rate and carrying capacity of strains evolved in monoculture provided little information on the relative abundance of strains in evolved communities.

This work addresses exciting questions about the degree to which evolution in a community context is repeatable and predictable. The experiment represents a tour de force with 87 communities and 16 monocultures all experimentally evolved with at least 3-fold replication. The manuscript provides interesting analyses demonstrating that even in a community context there is a degree of both repeatability and predictability to evolution.

With large data sets such as this there are many different analyses that could be done and no doubt the authors will continue to mine this data for many additional papers. There are however, a few analyses that I think would significantly enhance the current work.

First I would be interested to see data on the abundance change over the course of evolution for each species in monoculture, 2- and 3-species co-cultures. Much of this data is already present but typically presented as ratios, and not broken down by species. How do the distributions of change in co-cultures compare between species? Are there tight correlations in change across treatments for some species? These are clearly data that the authors have in hand so this should represent little additional work.

We are indeed performing additional analyses (and collecting more data) in the context of other questions regarding these data. But we agree that the analysis suggested by the reviewer can contribute to the current manuscript. We tried to follow the reviewer's advice, and performed the following analyses:

First, we broke down the data by species and compared their absolute abundance across different treatments - evolved in monoculture, evolved in a pair, and evolved in a trio. This analysis shows that absolute abundances often changed significantly throughout the experiment, and that the changes in abundances that occurred in pairs and trios tended to be more similar to each other than to those that occurred when species evolved in monoculture. We have included a new supplementary figure showing these results (Fig. S5, also copied below, referenced in the main text at lines 116, 233, and 235).

Another interesting point that this figure shows is that some species have a very strong trend in coculture. For example Ea seems to have a strong trend of drop in abundance during long-term coculturing with any other species. This raises an interesting question of what can explain this trend, and differences between species. We are very interested in this question, but answering it requires extensive further investigation, which we believe is outside the scope of this paper. Therefore we did not discuss this observation in the current manuscript.

Figure S5: Absolute abundance of species changes throughout the experiment. Absolute abundance is measured as the fraction of a species multiplied by the OD of the whole community. Blue and orange lines represent the median fractional OD of a species within all the pairs and trios respectively, and green line indicates the OD in monoculture. Shaded areas denote the .95 confidence interval.

In addition, we correlated the mean fold-increase in absolute abundance of species across treatments. There is a significant positive correlation between the mean increase in species' absolute abundance when evolved in pairs and in trios, but these do not correlate strongly with the species' increase during evolution in monoculture. This is consistent with the fact that pair composition is predictive of trio composition (Fig. 4) but monoculture growth is not predictive of coculture composition (Fig. 5). We have included a new supplementary figure showing these results (Fig. S5, also copied below).

Figure S10: Changes in abundance that occur when species evolve in pairs and in trios are correlated with each other, but not with those that occur when species evolve in monoculture. Fold-change in absolute abundance is measured as the log₂ ratio of the fractional OD (fraction of a species multiplied by the OD of the community) of a specific species in a specific replicate at generations ~400 and ~70. Values are averaged for each species in all its occurrences as a monoculture, in pairs, and in trios, separately. Histograms at the diagonal represent the distribution of average values for each one of the 16 species, in all communities with the same initial species' richness. Scatter plots compare the means of each species at two different community contexts. Error bars represent the standard deviation of a species' fold-change values for each treatment.

I would also be highly interested in the phenotypes of evolved isolates. How do growth rate and efficiency in monoculture change for populations evolved in monoculture and co-culture? This is relevant for example for thinking about the predictive value of r and K in the final section. These parameters for strains evolved in monoculture, do not predict the composition of evolved communities. However, it is unclear whether this lack of prediction is because populations evolve differently in monoculture and co-culture, or because growth rate and carrying capacity are poor predictors of community composition in the evolved communities. It may not be

feasible to determine growth rate and efficiency for all evolved isolates in monoculture, but even a representative subset would be informative.

How coevolution affects the phenotype of evolved strains is one of our major questions in the broader research that led to this manuscript. However, we still have only partial and preliminary data to answer this question. One thing we found for example (data not shown), is that coevolved strains almost always evolve lower growth capabilities than their mono-evolved counterparts. Given that this is still preliminary, and not directly related to predicting the community composition we plan to further investigate this in follow-up work, rather than in the context of the current manuscript.

We agree that correlating the growth abilities of coevolved strains is relevant for interpreting the fact that mono-evolved strains growth does not predict the evolved compositions. Therefore, we measured the carrying capacities of strains that were isolated from 21 pairs that coevolved for ~400 generations (Fig S13, attached below). Interestingly, these were not better than mono-evolved strains at predicting the evolved compositions. We did not measure growth rates for these strains since these measurements are much more laborious for such a large number of strains (126 unique strains, composed of 1-3 evolutionary replicates for each of the 21 pairs; Table S3).

Figure S13: Growth of strains that evolved in coculture do not predict the composition of pairs. The mean fraction of the species with the higher carrying capacity vs the ratio of the higher carrying capacity and the lower carrying capacity. Percentages indicate the percentage of pairs above the 0.5 line, which correspond to the prediction that the species with the higher growth rate/carrying capacity is more dominant in a pairwise competition. Data includes strains that were isolated from 1-3 evolutionary replicates of 21 pairs that evolved for ~400 generations (listed in Table S3)

We chose not to include these results in Figure 5 for since they are based on a significantly smaller number of (unique) pairs. Furthermore, in practice these results are less relevant for predicting community composition since such individual-growth data of coevolved strains is anyway not present or easily obtained.

However, these results had led us to change some of the interpretation we offered to the fact that growth of evolved strains did not predict the composition. Specifically, these results show that the lack of predictability of species's carrying capacities cannot be attributed to this trait evolving differently in monoculture and in coculture. An alternative possible explanation is given in line 273 :

“However, the growth ability of individual species was only predictive of community composition on ecological timescales (Fig. 5, S13). This suggests that when species coevolve, more complex interactions than competition for the supplied resources plays a larger role in determining community composition. For example, species may adapt to consume or to better tolerate their partners' secretions” (lines 269 - 274)

In the methods there are 2 null distributions described for determining repeatability, however it seems that only one comparison is presented in the paper. The supplement also has a comparison against a null distribution, but it appears it is the same comparison just with the data broken down into 2 and 3 species scenarios. Am I missing the second distribution?

We apologize for this inconsistency. In prior versions of the manuscript, we had another null model, which we decided to remove, as it did not add enough information and was hard to interpret. Therefore we took this null model out, but forgot to remove it from the methods section. It was removed from the methods section now.

Line 214 – I would encourage the authors to highlight that their measure of growth rate includes lag time. This is clearly explained in the methods, but I think it would help readers interpret results to mention it when the results are being discussed.

That is a good point. We added this in the caption of Fig. 5:

“Growth rate is measured as the time to a threshold of OD = 0.1 (Methods), and thus includes strains' lag time.” (lines 219-220)

Typo in line 295

Thanks for noticing, we fixed it.

Reviewers' Comments:

Reviewer #1:

Remarks to the Author:

This paper is very nice, I already liked the submitted version and I think it has been improved over it after the revisions, so I am supportive of its publication.

Alvaro Sanchez.

Reviewer #2:

Remarks to the Author:

I am satisfied with authors' response.

Reviewer #3:

Remarks to the Author:

I appreciate the authors' attention to my comments, and the comments of other reviewers. Specifically, I find breaking out the evolutionary dynamics by species useful. I have no further critiques.